# The Potential of Anti-Inflammatory DC Immunotherapy in Improving Proteinuria in Type 2 Diabetes Mellitus

**DOI:** 10.3390/vaccines12090972

**Published:** 2024-08-27

**Authors:** Jonny Jonny, Enda Cindylosa Sitepu, I Nyoman Ehrich Lister, Linda Chiuman, Terawan Agus Putranto

**Affiliations:** 1Indonesia Army Cellcure Center, Gatot Soebroto Central Army Hospital, Jakarta 10410, Indonesia; endacsitepu@gmail.com (E.C.S.);; 2Faculty of Medicine, Dentistry, and Health Sciences, University Prima Indonesia, Medan 20118, Indonesia; 3Faculty of Military Medicine, Indonesia Defense University, Jakarta 16810, Indonesia; 4Faculty of Medicine, University of Pembangunan Nasional “Veteran” Jakarta, Jakarta 12450, Indonesia

**Keywords:** dendritic cells, diabetic kidney disease, cell immunotherapy, proteinuria, diabetes mellitus

## Abstract

A typical consequence of type 2 diabetes mellitus, diabetic kidney disease (DKD) is a significant risk factor for end-stage renal disease. The pathophysiology of diabetic kidney disease (DKD) is mainly associated with the immune system, which involves adhesion molecules and growth factors disruption, excessive expression of inflammatory mediators, decreased levels of anti-inflammatory mediators, and immune cell infiltration in the kidney. Dendritic cells are professional antigen-presenting cells acting as a bridge connecting innate and adaptive immune responses. The anti-inflammatory subset of DCs is also capable of modulating inflammation. Autologous anti-inflammatory dendritic cells can be made by in vitro differentiation of peripheral blood monocytes and utilized as a cell-based therapy. Treatment with anti-inflammatory cytokines, immunosuppressants, and substances derived from pathogens can induce tolerogenic or anti-inflammatory features in ex vivo–generated DCs. It has been established that targeting inflammation can alleviate the progression of DKD. Recent studies have focused on the potential of dendritic cell–based therapies to modulate immune responses favorably. By inducing a tolerogenic phenotype in dendritic cells, it is possible to decrease the inflammatory response and subsequent kidney damage. This article highlights the possibility of using anti-inflammatory DCs as a cell-based therapy for DKD through its role in controlling inflammation.

## 1. Introduction

Twenty to fifty percent of diabetic people develop diabetic kidney disease (DKD), which is a significant risk factor for end-stage kidney disease [1]. Over the course of the year, more than two million people are newly diagnosed with DKD globally, and this figure is increasing [2]. The management of DKD cases represents a significant financial burden. This disease also significantly decreases the quality of life. Therefore, developing effective DKD therapies that can minimize side effects is necessary.

DKD is characterized by persistent proteinuria, gradual decline in renal function, and histologically appears as a glomerular disease [3]. In DKD, structural and functional changes occur in the kidneys [4]. Mesangial enlargement in the glomerulus, basement membrane thickening, loss of podocytes, nodular glomerulosclerosis, and damage to endothelial cells are among the pathological characteristics of DKD [5]. Tubular hypertrophy in the early stages of DKD might progress to fibrosis in the interstitial with atrophy of tubules [6].

In DKD, there is also an increase in albumin excretion and disruption of glomerular filtration [7]. However, there is also a nonclassical presentation where DKD occurs without albuminuria. Non-albuminuric DKD (NA-DKD) is hypothesized to be caused by macroangiopathy leading to interstitial fibrosis and vascular lesions. In NA-DKD, interstitial fibrosis due to this macroangiopathy is not associated (independent) with albuminuria [8,9]. Some studies suggest that renal impairment in NA-DKD is not caused by hyperglycemia or microangiopathy but rather by genetic susceptibility, aging, and arteriosclerosis [10]. This article will focus on the classical presentation of DKD with albuminuria due to these differences in pathogenic mechanisms.

Dendritic cells (DCs) are innate immune cells that function as antigen-presenting cells. They are considered the “master regulator” of immunity. DCs connect innate and adaptive immunity. DCs express MHC class I and MHC class II molecules, allowing the presentation of antigens to both subsets of CD4+ T cells and CD8+ T cells (cross-presentation). The immune response triggered by DCs is diverse. Some antigens presented by DCs evoke an immunogenic response, activating the immune system to fight against the antigen. However, another subset of DCs that have tolerogenic phenotypes also plays a role in stimulating anti-inflammatory responses, thereby limiting excessive inflammation [11].

Type 2 diabetes mellitus (T2DM) involves metabolic abnormalities that lead to chronic low-grade inflammation. This chronic inflammation ultimately results in various complications, such as DKD. Administering agents that can control inflammation in T2DM patients has the potential to serve as therapy as well as prevention for DKD. Therefore, this article aims to elucidate the biomechanisms that underlie the development of DC-based DKD therapy.

## 2. Diabetic Kidney Disease Immunopathology

The immunopathological aspects of type 2 diabetes mellitus (T2DM) constitute a complex mechanism. Immune dysregulation in T2DM is caused by the interaction of genetic, immunological, metabolic, and clinical elements [12]. Although type 2 diabetes mellitus (T2DM) is generally classified as a nonautoimmune disease, recent findings suggest the involvement of autoimmune processes in this condition [13,14]. However, inflammation, insulin resistance, and a decrease in pancreatic beta cells are the leading causes of type 2 diabetes mellitus (T2DM) [14]. Understanding the immunological processes and conditions affected by the immune system is essential in developing an effective therapy for DKD.

Chronic inflammation is a primary immune system-related pathology that plays a vital role in developing T2DM disease [15,16,17,18]. Inflammation causes various microvascular and macrovascular complications in diabetics [17]. T2DM is associated with activated inflammatory signaling and abnormal cytokine production [19]. Prolonged mild inflammation and elevated proinflammatory markers are closely associated with the development and progression of T2DM [17]. Inflammation also causes insulin resistance, beta cell dysfunction, and complications of diabetes [20,21].

Studies have shown that inflammation, characterized by increased C-reactive protein (CRP), is closely related to insulin resistance [22]. In addition, obesity provokes chronic inflammation, contributing to the formation of insulin resistance [20,23]. Subsequent research has revealed a connection between the resistance to insulin and cardiovascular diseases in T2DM patients with coronary heart disease, which is caused by systemic inflammation mediated by high-sensitivity CRP (hsCRP) [24]. Inflammation not only plays a role in the pathogenesis of T2DM but also its complications, including nonalcoholic fatty liver, retinopathy, DKD, and cardiovascular problems [20]. Chronic inflammation is strongly associated with insulin resistance in individuals with T2DM, so addressing inflammation is critical in the management and prevention of T2DM [22,23,24].

The chronic inflammatory condition in T2DM causes various immune disorders that contribute to the pathogenesis of DKD. Molecularly involved factors start from transcription factors such as Nuclear Factor Kappa B (NF-κB) [25,26,27], Janus kinase/signal transducers and activators of transcription (JAK/STAT) [28,29], and adenosine monophosphate–activated protein kinase (AMPK) [28,29,30]. In addition, infiltration of immune cells in the kidneys also affects the course of DKD disease. Cytokines, chemokines, and related adhesion molecules determine the fate and reactivity of immune and nonimmune cells in the kidneys [31,32,33]. The accumulation of all these factors causes abnormalities in the renal glomerulus, resulting in DKD.

In DM, glucose and metabolites activate macrophages present in the kidneys [34]. Macrophages release cytokines, recruit peripheral monocytes/macrophages, and increase kidney cell injury, ultimately resulting in inflammation and fibrosis [35]. Macrophages can be activated into proinflammatory phenotypes by pathogen-associated molecules released from injured kidney cells, stimulating the recruitment of other inflammatory cells and activating renal fibroblasts [36]. In chronic kidney disease, persistent activation of proinflammatory monocytes and persistence of reparative macrophages contribute to glomerulosclerosis and tubulointerstitial fibrosis [35]. Macrophage accumulation in the kidneys is associated with the occurrence of glomerulonephritis [37]. Autophagy and lysosome degradation pathways also contribute to macrophage polarization, chronic inflammation, and organ fibrosis [38]. Regulation of autophagy also plays a role in developing glomerulus-related diseases [39]. This event ultimately triggers the emergence of proteinuria in DM patients.

Several studies have shown T cells’ pathogenic role in the induction of proteinuria in DKD [40,41,42]. Activation of innate cells from the immune response and release of inflammatory cytokines like interferon-γ (IFN-γ) and tumor necrosis factor-α (TNF-α) are linked to an increase in T cells in the circulation and renal cortex [43]. Once circulating, T cells are recruited into renal tissue or amplified, differentiated, and activated in the renal. These T cells then mediate various pathogenic mechanisms, such as influencing insulin resistance, mediating podocyte damage, inducing fibrosis, and regulating proteinuria [33].

In DM, there is a disruption in the balance of inflammatory mediators (IL-6, TNF-α, TGF-β) and anti-inflammatory mediators such as IL-10. Hyperglycemia conditions trigger resident kidney cells (mesangial and podocyte cells), interstitial tissue, and tubules to produce IL-6 [44]. JAK/STAT, which mainly triggers cell proliferation, is continuously activated, resulting in hypertrophy of podocytes [45,46]. Furthermore, through activation of the rapamycin complex, podocyte hypertrophy continuously occurs and ultimately leads to the release of podocytes from the basal glomerular membrane (MBG) (due to changes in the structure of podocytes) [47] and apoptosis podosit [48]. IL-6 has also been shown to play a role in mesangial expansion, as evidenced by the discovery of IL-6 in the mesangium, interstitium, and renal tubules [33]. This IL-6 display also triggers the release of MCP-1 by mesangial cells, thus increasing monocyte recruits [49]. Podocyte hypertrophy, podocyte cell loss, and mesangial expansion impaired the glomerular function characterized by proteinuria and decreased the glomerular filtration rate (GFR) [50].

TNF-α is produced by macrophages/monocytes that infiltrate the kidneys. Still, mesangial cells, podocytes, and tubular epithelium can also release TNF-α after being stimulated by hyperglycemia and advanced glycation end products (AGEs) [51]. TNF-α works by activating various secondary proteins that cause activation of gene transcription and production of reactive oxygen species or nitrogen radicals (NO). TNF-α can activate G-proteins, transcription factors (e.g., NF-kB, AP-1), protein kinases (e.g., CK-II, erk-1, erk-2, and MAP2), phospholipases, mitochondrial proteins, and proteases [52]. In DKD pathogenesis, TNF-α also increases the expression of adhesion and chemokine molecules, thus aggravating renal microinflammatory conditions. TNF-α also increases renal cell cytotoxicity/necrosis by inducing ROS and NO. In podocytes, TNF-α induces cytoskeleton reorganization and decreases cell viability. In addition, TNF-α also causes changes in intraglomerular blood flow and GFR and increases endothelial permeability [51].

Hyperglycemia leads to increased glucose transport-1 (GLUT-1) regulation, resulting in excess TGF-β expression by renal mesangial and tubular cells. Moreover, increased intraglomerular pressure, stretching of mesangial cells, activation of the renin-angiotensin system, ROS, and advanced glycation end products (AGEs) induce TGF-β production in renal mesangial and tubular cells [53]. TGF-β1 functions as a profibrotic mediator in various kidney diseases [54] and also acts on renal mesangial cells and fibroblasts by inducing cell proliferation, cell migration, and transcription of profibrosis molecules (collagen, fibronectin, and plasminogen activator inhibitor-1 (PAI-1)). The mechanism of indirect fibrosis by TGF-β1 is still not widely studied, but Das et al. show that TGF-β1 initiates an apoptotic cascade in podocytes, leading to podocyte loss [55].

Interleukin 10 (IL-10) was initially known as an inhibitory factor of cytokine synthesis [56]. Activation of immunity and chronic inflammation contribute to the pathogenesis of T2DM and its progression to DKD [57]. Patients with T2DM have lower IL-10 levels [58]. This is exacerbated by the resistance of immune cells to the anti-inflammatory effects of IL-10 [59]. IL-10 has the potential to slow progression and improve the prognosis of DKD. In DKD, the accumulation of inflammatory cells (leukocytes, monocytes, and macrophages) in the kidneys synthesizes more proinflammatory and fibrogenic cytokines that damage kidney structure directly [60]. IL-10 can suppress this through its anti-inflammatory properties by reducing inflammatory cell infiltration, reducing renal interstitial fibrosis, and preventing mesangial cell expansion [61]. In vitro, it was found that IL-10 can reduce reactive oxygen species (ROS) levels and, in embryonic mouse fibroblast cells, can reduce collagen synthesis [62]. Directly, IL-10 reduces glomerular macrophages’ recruitment, activation, and proliferation in vivo. In addition, IL-10 also significantly reduces macrophage-mediated glomerular injury and improves proteinuria conditions [62].

In the kidneys, this adhesion molecule also plays a role in the attachment of leukocytes to the endothelium, promotes macrophage accumulation, increases TGF protein synthesis by tubular cells, and increases adhesion and activation of T cells [63]. Molecular adhesion can be induced by the following conditions commonly found in type 2 DM patients: hyperglycemia, advanced glycation end product (AGE), oxidative stress, and hyperinsulinemia [63]. Research on experimental animal models found a causative relationship between VCAM ICAM and DKD. Okada et al. found that mice in ICAM-1-deficient diabetes models had higher rates of macrophage infiltration in the kidneys than controls. Nephropathy can be reduced by inhibiting ICAM-1 expression [64]. Chow et al. also found that ICAM-1 deficiency can reduce macrophage accumulation in the glomerulus to decrease glomerular hypertrophy and interstitial fibrosis [65]. Thus, therapeutic modalities targeting decreased VCAM-1 and ICAM-1 expression in the kidney could be DKD therapy [63].

Podocytes and tubule cells will produce VEGF-A continuously, unlike other tissues that stop making it when blood vessel development is complete. In the kidneys, VEGF plays a vital role in angiogenesis, endothelial cell proliferation and survival, and regeneration of damaged tissues. Hyperglycemia conditions will trigger a chain reaction to cause an accumulation of VEGF-A and ultimately cause microvascular complications of DM. The renin–angiotensin system, ROS, and AGEs64 influence this chain reaction. Angiotensin II will increase the production of VEGF-A and vascular inflammation [66]. VEGF-A expression increases as ROS production increases [67]. Meanwhile, AGEs can cause mesangial cells to apoptosis and increase VEGF expression, causing increased vascular permeability [68]. VEGF in the kidneys is mainly sourced from podocytes and tubular cells. Therefore, in the early phases of kidney damage, VEGF increases. However, severe kidney damage found a decrease in VEGF as a result of damage so extensive that cells in the kidneys were no longer able to express VEGF [69].

Matrix metalloproteinase (MMP) also plays an important role in developing DKD disease. MMP-9, or gelatinase B, is an endopeptidase that causes the degradation of extracellular matrix proteins such as collagen, fibronectin, and laminin. The main substrate of MMP-9 is type 4 collagen; in the kidney, type 4 collagen is the main structure that makes up the glomerular filtration barrier, especially the glomerular basement membrane [70]. Neutrophils, macrophages, and fibroblasts70 mostly express MMP-9 [71]. However, in diabetic kidney injury, there is also an increase in MMP-9 expression in proximal renal tubule epithelial cells [72]. An increase in MMP-9 in the kidneys leads to an increase in various activities of chemokine chemokines (CCXCL5, CXCL8), cytokines (TNF-α, IL1β, TGF-β), receptors, growth factors, and other cytokine adhesion molecules. On the other hand, MMP-9 can also inactivate CXCL1, CXCL4, CXCL5, CXCL7, CXCL12, and IL1 β [71]. MMP-9 activity causes endothelial–mesenchymal transition, tubulointerstitial, and inflammatory fibrosis [73].

Cytotoxic T-lymphocyte-associated protein 4 (CTLA-4) is an inhibitory receptor of the immunoglobulin CD28 subfamily. CTLA-4 is an essential molecule for helper T cells and podocytes. In hyperglycemia, there is an increase in CD28, which is a marker of podocyte damage. This increase in CD28 mediates T cell infiltration, thus aggravating podocyte damage. By acting as a negative regulator of T cell activation, CTLA-4 guards against harm to podocytes. Research shows that CD28/B7/CTLA-4 polymorphisms increase susceptibility to DKD in Type 2 diabetes patients in China [74]. Inhibition of CTLA-4 has become one of the focuses of DKD therapy development [75,76,77].

Furthermore, the cyclic GMP-AMP synthase (cGAS) and stimulator of interferon genes (STING) also play a significant role in the immunopathology of DKD. This pathway is a critical part of the innate immune system, designed to detect cytosolic DNA and trigger an immune response [78]. The pathway is initiated when cGAS recognizes and binds to cytosolic double-stranded DNA (dsDNA) originating from pathogens like viruses or damaged host cells. Upon binding to DNA, cGAS undergoes a conformational change that allows it to synthesize cyclic GMP-AMP (cGAMP) from ATP and GTP. cGMP acts as a second messenger, binding to and activating STING, which is localized on the endoplasmic reticulum (ER) membrane. Activation of STING induces its translocation from the ER to the Golgi apparatus, where it recruits TANK-binding kinase 1 (TBK1). TBK1 then phosphorylates STING and the transcription factor IRF3 (Interferon Regulatory Factor 3). Phosphorylated IRF3 dimerizes and translocates to the nucleus, promoting the transcription of type I interferons and other inflammatory cytokines. In parallel, STING activation also triggers the NF-κB signaling pathway, further producing proinflammatory cytokines [79]. The combined action of IRF3 and NF-κB results in a robust immune response to clearing intracellular infections and managing cellular stress.

This pathway also contributes to chronic inflammation observed in metabolic disorders such as diabetic kidney disease (DKD), contributing to ongoing inflammation and tissue damage [80]. In diabetic conditions, high glucose levels induce oxidative stress within cells, leading to mitochondrial damage. This damage disrupts the mitochondrial membrane, causing the leakage of mtDNA into the cytoplasm and eventually into the circulation [81]. mtDNA damage and subsequent leakage into the cytosol trigger the activation of the cGAS-STING pathway. This activation leads to an inflammatory cascade, marked by the production of proinflammatory cytokines such as TNF-α and IL-6, contributing to podocyte injury and the breakdown of the glomerular filtration barrier, exacerbating albuminuria and renal dysfunction [80].

The cGAS-STING pathway also plays a significant role in dendritic cell (DC) function. Activation of this pathway, primarily through the sensing of cytosolic DNA, leads to the production of type I interferons (IFN-I), crucial for DC activation and subsequent immune responses [82]. In metabolic diseases, excessive activation of the cGAS-STING pathway can trigger chronic inflammation, autophagy, and apoptosis. Moreover, metabolic stressors, including mitochondrial and nuclear DNA damage, can further activate this pathway, exacerbating inflammation and metabolic dysfunction [81]. Inhibition of the STING pathway leads to a lower activation of DC, which is shown to have a protective effect against DKD progression [83,84,85].

## 3. Kidney Dendritic Cell Subsets

A study has revealed four subsets of mononuclear phagocytes (MPs) in the adult kidney, all characterized by a prominent Clec9a-expression history. These subsets include conventional dendritic cell type 1 (cDC1), which plays a crucial role in immune responses, and conventional dendritic cell type 2 (cDC2), another critical dendritic cell subset with distinct functions. Additionally, a subgroup of CD64-expressing CD11b^hi^ cells, marked by their expression of CD64 and CD11b, indicates a specific functional role within the kidney’s immune landscape. Furthermore, F480^hi^ cells are identified as a subset with distinct properties, phenotypically similar to macrophages. These four populations are shown to be phenotypically, functionally, and transcriptionally distinct [86].

The cDC1 cells, which express CD103 and are located near blood vessels, play a critical role in activating T cells and are essential for adaptive immunity. In mice, these cells do not express SiglecH or Ly6C. Most dendritic cells in the kidney fall under the cDC2 category, characterized by CD11b and CX3CR1 expression. They are primarily found in the kidney cortex, where they participate in various immune responses [87]. Kidney-specific dendritic cells are also noted for their ability to migrate in response to chemokines detected by receptors on precursor dendritic cells. Kidney DCs express chemokine receptors like CCR1, CCR2, CCR5, CCR7, and CX3CR1, with CCR5 facilitating their entry into a healthy kidney. In the case of inflammation, monocyte-derived DCs, which differentiate from peripheral blood monocytes, migrate to the injured kidney through the expression of CCR2 and CX3CR1 [88]. Once activated, the migration of kidney DCs to the kidney-draining lymph nodes is driven by CCR7 [89].

In normal renal tissue, kidney dendritic cells (DCs) form an intricate network around the tubules, interstitium, and glomeruli, constantly monitoring their surroundings. When encountering self-antigens or minor molecular weight antigens from the glomerular filter, they remain immature and promote immune tolerance by either inducing apoptosis in autoreactive T cells or converting T helper cells into regulatory T cells [90]. In this steady state, kidney DCs express low-costimulatory molecules like CD80 and CD86, indicating their immature status and role in suppressing adaptive immune responses. Kidney DCs detect pathogens during acute infections and initiate host defense by activating innate immune cells such as granulocytes and macrophages, leading to inflammation and kidney damage [91].

An increase in DCs within the kidney interstitium happens in chronic kidney disease (CKD), where they contribute to disease progression, even in nonimmune-driven conditions like hypertensive or obstructive nephropathy. However, the impact of different DC subsets on kidney injury varies, as shown by variable outcomes in studies depleting the entire DC population [92]. Human kidney biopsy studies reveal higher numbers of total DCs, particularly CD141^hi^ and CD1c^+^ conventional DC subsets (cDC1s and cDC2s), in diseased kidneys with interstitial fibrosis compared to nonfibrotic or healthy tissues [93,94]. This suggests that activated cDCs play a crucial role in fibrosis development and CKD progression. Studies also indicate that angiotensin II in nonimmune renal disease models leads to DC accumulation and maturation; therefore, DCs are implicating kidney DCs in tubulointerstitial damage in diabetic nephropathy (DN) [95]. While the role of DC subsets in DN is not fully understood, evidence suggests that cDC1s may play a significant role in disease progression, making them potential targets for DN treatment.

## 4. The Role of DCs in Inducing an Anti-Inflammatory Response

DCs are the primary regulators of the immune system. The role of dendritic cells in the immune system is that they function as a response controller to pathogens and are actively involved in inflammatory resolution [96]. Various studies have successfully revealed the anti-inflammatory properties of DCs, which further adds to the complexity of DCs’ role in immunity [97,98,99,100]. Generally, DCs play a role in antigen presentation and T cell activation. However, DCs have also been known to play a central role in modulating inflammatory responses. DCs relieve inflammation, prevent excessive tissue damage, and promote immune homeostasis [100].

One of the critical anti-inflammatory mechanisms of DCs is the secretion of anti-inflammatory cytokines. Anti-inflammatory DCs can produce cytokines such as interleukin-10 (IL-10) and transforming growth factor-beta (TGF-β), which are immunosuppressive. These cytokines act on other immune cells, such as T cells and macrophages, to dampen the proinflammatory response and increase inflammatory resolution [98,101]. Dendritic cells are essential in promoting differentiation and activation of regulatory T cells (Tregs), a specialized subset of T cells with immunosuppressive solid functions. DCs create an environment conducive to Treg development by presenting antigens through tolerogenic and regulatory cytokine secretion. The Treg cell then directly suppresses the exaggerated immune response and contributes to the resolution of inflammation [99].

Under certain conditions, dendritic cells undergo phenotypic changes, becoming tolerogenic or anti-inflammatory DCs characterized by a reduced expression of co-stimulating molecules and an increased production of anti-inflammatory cytokines. Anti-inflammatory DCs actively provoke the formation of an anti-inflammatory microenvironment, influence T cell development into anti-inflammatory phenotypes, and contribute to the resolution of inflammation [102].

Anti-inflammatory DCs can actively suppress effector T cell responses, thereby preventing inflammatory reactions. Through the induction of T cell anergy or by promoting T cell apoptosis, DCs contribute to the downregulation of immune responses and inflammatory resolution [100]. Dendritic cells also affect macrophage polarization by directing macrophages into the M2 phenotype, which is anti-inflammatory. This modulation of macrophage function contributes to tissue repair and inflammatory resolution since M2 macrophages are involved in tissue remodeling and the clearance of cellular debris [103]. DCs engage in negative feedback regulation to control the intensity and duration of the immune response. By expressing inhibitory receptors (e.g., CTLA-4, LAG-3, PD-1) and interacting with regulatory molecules, DCs modulate their activation and prevent excessive inflammation, thereby restoring immune homeostasis [102].

Understanding DC-mediated anti-inflammatory mechanisms opens up exciting possibilities for development as therapeutic interventions in inflammatory disorders and autoimmune diseases. Utilization of Treg cells immunoregulator properties induced by DCs may be one of the therapeutic interventions. Strategies to increase Treg activity or trigger its expansion may also be a way to elicit an anti-inflammatory response. Innovations in precision medicine enable the development of personalized therapies based on individual immune profiles. Thus, interventions modulating DC function in patients can optimize therapeutic outcomes in inflammatory disorders.

## 5. Ex Vivo Production of Autologous Dendritic Cells

Two approaches can make DCs ex vivo: MoDC differentiation from peripheral blood mononuclear cells (PBMC) with a mixture of maturation cytokines and DC differentiation from CD34+ bone marrow precursors, which are hematopoietic stem cells with multipotent capabilities. PBMC stimulated with a mixture of maturation cytokines will produce a subset of MoDC. In contrast, differentiation of the CD34+ bone marrow precursor produces a mixture of a subset of MoDC, DC, which phenotypically resembles Langerhans cells (DC present in the epidermis), and myeloid cells at various levels of maturation [104,105]. Although much evidence suggests that cDC is better at cross-presenting exogenous antigens to MHC-1, cDC production is still quite difficult [106]. MoDC is the most widely available and versatile subtype of DC104 in humans [107]. MoDC was created ex vivo from PBMC, which is still the most frequently used method in developing DC-based therapies, both cancer and infection therapies. MoDC is more widely used not because of its superior clinical efficacy but because it is more widely available than CD34+ precursors in blood taken by apheresis [105]. The isolation of CD34+ from the bone marrow through bone marrow punctuation is an invasive procedure with a high risk. Thus, MoDC made from PBMC is the best choice today for manufacturing DCs for therapeutic purposes.

PMBC is differentiated into DC immature by culturing cells for 5–7 days along with GM-CSF (granulocyte-macrophage colony-stimulating factor) and Interleukin-4 (IL-4) [107]. GM-CSF is a cytokine that stimulates the differentiation of hematopoietic stem cells into various types of cells, including monocytes in PBMC [108]. GM-CSF is a cytokine that stimulates the differentiation of hematopoietic stem cells in multiple kinds of cells, including monocytes present in PBMC [109]. Thus, the mixture of the two cytokines synergistically promotes the differentiation of monocytes into MoDC and their maturation into APC.

Anti-inflammatory DCs can be generated by culturing MoDC and specific agents such as Vitamin D3 and Dexamethasone. Maturation stimuli such as LPS are given so that semi-mature anti-inflammatory DCs are obtained [110]. MoDCs cultured with probiotic bacteria such as *L. delbruekii* and *L. rhamnosus* also exhibited tolerogenic phenotypes [111]. Another method that can be performed is to induce a tolerogenic phenotype in DCs through transfection [112]. Interestingly, DCs made from bone marrow cultures and GM-CSF and IL-4 without being given an antigen stimulus (DC immature) also showed the ability to control inflammation [113]. Thus, DC transfer without antigen stimulus can be considered for administration as therapy in diseases based on inflammatory disorders.

## 6. Current State of Cell-Based Therapy for DKD

Autologous DC transfer is commonly used as a therapy for cancer, chronic infections, autoimmune diseases, and infection-preventing vaccines [114]. The use of autologous DC transfer in metabolic and degenerative disease therapy has yet to be carried out. However, there are other cell-based therapies to improve conditions in metabolic diseases that have been studied. This section will discuss some significant findings related to cell therapy performed in patients with DKD.

A study by Dubsky et al. found that systemic and local administration of autologous mononuclear cells in the form of hematologic stem cells in DM subjects with chronic kidney disease (CKD) can protect against amputation due to critical limb ischemia, including patients with end-stage renal disease [115]. Critical limb ischemia is caused by severe occlusion of the arteries of the lower extremities (peripheral artery disease) and is one of the vascular complications of DM. Hyperglycemia in DM triggers the formation of advanced glycation end products (AGEs), which then increase the uptake of oxidized low-density lipoproteins by macrophages and then develop into foam cells. Foam cells accumulate in the subendothelial area of the artery wall, forming atherosclerosis lesions [116]. Atherosclerosis in this artery then blocks blood flow to the distal region, resulting in critical limb ischemia.

Peripheral artery disease (PAD) generally occurs in the large arteries, so it is classified as macroangiopathy. However, the fact is that PAD is also accompanied by local and systemic microangiopathy [117]. PAD is closely related to DKD. This is due to endothelial dysfunction in various blood vessels, causing thickening of the basement membrane of capillary arteries, endothelial hyperplasia, decreased oxygen pressure, and hypoxia, causing disorders of multiple organs, including the kidneys. In addition, reduced kidney function is also one factor supporting the severity of PAD—AGEs are factors that trigger the formation of physiological foam cells that are eliminated by the kidneys. Thus, along with the decline in kidney function, the number of AGEs will increase, ultimately increasing the possibility of critical limb ischemia.

Dubsky et al. conducted experiments by giving mononuclear cell transfer CD34+ (bone marrow stem cell) to patients with end-stage renal disease, which proved to protect against amputation. The therapeutic effects of CD34+ cell transfer are due to the immunomodulating impacts through direct contact with innate and adaptive immune cells and paracrine effects caused by the production of cytokines, chemokines, and growth factors [118]. Although the study did not see any impact on the kidneys, given that PAD in DM patients is closely related to kidney function, the protective effect of critical limb ischemia may also be influenced by the renoprotective effect through modulating the immune system systemically after CD34+ mononuclear cell transfer.

Some researchers have also conducted autologous cell transfer experiments as DKD therapy. A systematic evaluation and meta-analysis revealed that stem cell treatment can enhance kidney function in experimental animals with DKD. Reductions in proinflammatory markers (TNF-α, IFN-γ, IL-6, IL-8, MCP-1), reductions in fibrosis indicators (TGF-β, Collagen I, and IV, fibronectin), and increases in anti-inflammatory markers (IL-10) were linked to improvements in kidney function [119]. However, human testing has not shown satisfactory results. A randomized placebo-controlled clinical trial on allogeneic mesenchymal precursor cells (MPCs) administration in DKD patients found that there was no significant difference in the ratio of urinary creatinine–albumin, serum creatinine, and HbA1c in the test group compared to placebo [120].

Stem cells are hypothesized to improve pathology in metabolic diseases because stem cells’ multipotential ability is expected to regenerate damaged cells. However, in the context of DKD therapy, the mechanism of action of cell-based therapy is mainly based on the ability to modulate immunity and anti-inflammatory effects of stem cells because there is not enough evidence to suggest the engraftment of systemically administered stem cells [120]. The use of stem cells for DKD has several disadvantages, namely the presence of metabolic memory, which causes stem cells obtained from DKD patients to experience decreased multipotency and immunomodulatory abilities, are more susceptible to apoptosis, and experience increased senescence. In addition, there is a risk of teratoma formation from stem cells [121].

Overall, the current research shows that cell-based therapy, such as DKD, can be applied as one of the approaches to metabolic disease therapy. Current research focuses on developing stem cell-based therapies. Still, current evidence suggests that the mechanism of action of stem cells as DKD therapy emphasizes paracrine immunomodulating and anti-inflammatory abilities. Thus, other cell-based products with similar capabilities also have the potential to be developed.

## 7. Current Treatment of DKD and Its Effect on the Immune System

The management of diabetic kidney disease (DKD) has evolved significantly, with several pharmacological interventions demonstrating an effectiveness in slowing disease progression. Critical treatments include renin–angiotensin–aldosterone system (RAAS) inhibitors, such as angiotensin-converting enzyme (ACE) inhibitors and angiotensin receptor blockers (ARBs), which have established roles in reducing proteinuria and hypertension associated with DKD [122,123,124]. Recent advancements highlight the importance of sodium-glucose cotransporter-2 inhibitors (SGLT2i) and glucagon-like peptide-1 receptor agonists (GLP-1 RAs), which have shown promising renal protective effects beyond glycemic control [122,125].

ACE inhibitors are known to reduce proteinuria and indirectly affect the immune system. The beneficial effects of an ACE-inhibitor on renal function may be due to its ability to block the renin–angiotensin system (RAS), which in turn reduces the tubular production of monocyte chemoattractant protein-1 (MCP-1) [126]. This suppression of MCP-1 is crucial because it leads to decreased monocyte infiltration and interstitial fibrosis in the kidneys, which are critical factors in the progression of diabetic nephropathy. ACE inhibitors may also reduce oxidative stress, closely linked to inflammation [127]. By decreasing oxidative stress, these medications can help lower the inflammatory response in various tissues, including the kidneys [127]. ACE inhibitors can also restore the number of leukocytes depleted in patients with diabetic kidney disease. However, they also tend to increase T cell infiltration in the glomerulus and polarize macrophage toward the M1 phenotype [126]. M1 macrophages are known to have inflammatory functions and mainly mediate fibrosis by secretion of TGF-β. This indicates that ACE inhibitors should be used cautiously because they might worsen the fibrosis in DKD.

SGLT2i show promising results in decreasing proteinuria. They also affect the immune system by modulating inflammation by reducing proinflammatory cytokines, modulating immune cell activity, improving metabolic parameters, promoting renal protection, and reducing oxidative stress [128]. SGLT2i are found to activate the energy sensor complex AMPK/SIRT1 and downregulate oxidative stress in mouse models [129]. SGLT2i also upregulates complement receptor type 1-related protein y (Crry), an essential complement regulator inhibiting complement overactivation, accompanied by suppressing hypoxia factor HIF-1α, which reduces inflammation in the kidney [130]. SGLT2 inhibitors also contribute to lowering levels of proinflammatory mediators, such as IL-6 and TNF-α, and help reduce body mass over time, which may further support kidney health [131,132]. SGLT2 inhibitors have also been documented to attenuate the activation of the NLRP3 inflammasome [133]. NLRP3 activation is linked to the progression of kidney damage in diabetic patients. Thus, attenuation of NLRP3 reduces the inflammatory response within the kidney, creating a more favorable environment for kidney health.

Glucagon-like peptide-1 receptor agonists (GLP-1 RAs) have been shown to exert beneficial effects on the immune system in DKD. Research indicates that GLP-1 RAs, such as liraglutide, can reduce inflammation by downregulating the receptor for advanced glycation end products (RAGE), which is implicated in proinflammatory pathways associated with DKD [134]. This downregulation leads to a decrease in the expansion of myeloid progenitors and promotes M2-like macrophage polarization, which is related to anti-inflammatory responses [134]. Additionally, GLP-1 RAs exhibit antioxidant properties and can mitigate oxidative stress, further contributing to their kidney-protective effects [135].

Overall, clinical trials have shown that ACE-I, SGLT2i, and GLP-1 RAs have beneficial effects in lowering proteinuria in DKD patients. These drugs also act on the immune system by reducing inflammation, which in turn protects the kidneys from immune-mediated damage.

## 8. Anti-Inflammatory DC Mechanism of Action in DKD

DC immunotherapy is created by modifying monocytes into DCs by stimulating differentiated cytokines. The product is a cell with a late differentiation stage, so teratoma formation is not risky. However, as previously explained, DCs have immunomodulatory abilities through direct interaction with other immune cells and the paracrine effect of released cytokines and growth factors. In addition, DCs are components of the immune system whose primary physiological function is to regulate the immune response. Thus, the immunomodulation function of DCs is likely better than that of stem cells. Therefore, the effectiveness and mechanism of action of DC transfer as DKD therapy need further investigation.

Based on existing research, it was found that DCs can induce an anti-inflammatory response [136]. Through the production of anti-inflammatory cytokines, DCs can systemically inhibit low-grade chronic inflammation. In addition, DCs that express fewer co-stimulating molecules on the surface also directly inhibit the activation and anergy of kidney-infiltrating T cells. DCs with tolerogenic phenotypes also express inhibitory molecules such as CTLA-4 on their surfaces to directly inhibit T cell function [137]. DCs are also known to support the proliferation of macrophages into the anti-inflammatory M2 type. In addition, DCs can stimulate the proliferation of peripheral Treg cells that firmly control the immune response [52].

Chronic low-grade inflammation in DM contributes to fibrosis, endothelial dysfunction, and hypoperfusion underlying DKD pathology [138,139]. Thus, DC therapy should be given systemically so that systemic immune immunomodulation occurs. This is expected to stop further immune cell–mediated damage to the kidneys and maintain kidney function (Figure 1). However, DKD is a progressive chronic disease. Products that can induce long-term anti-inflammatories are needed to provide clinically significant effects. Therefore, it is necessary to know the dose (number of cells), frequency, and appropriate period of administration so that a satisfactory clinical effect can be achieved.

## 9. Conclusions

DKD is caused by kidney damage as a result of chronic low-grade inflammation in type 2 diabetes patients. Inflammation causes damage to kidney structure and function, characterized by proteinuria and progressive decline in kidney function. Anti-inflammatory DCs can induce a robust anti-inflammatory response through the secretion of anti-inflammatory cytokines, inhibit activation, give rise to T cell anergy, and support M2 proliferation and Treg cell stimulation. Thus, DC transfer in DKD patients has the potential to be a new therapeutic approach.

## Figures and Tables

**Figure 1 vaccines-12-00972-f001:**
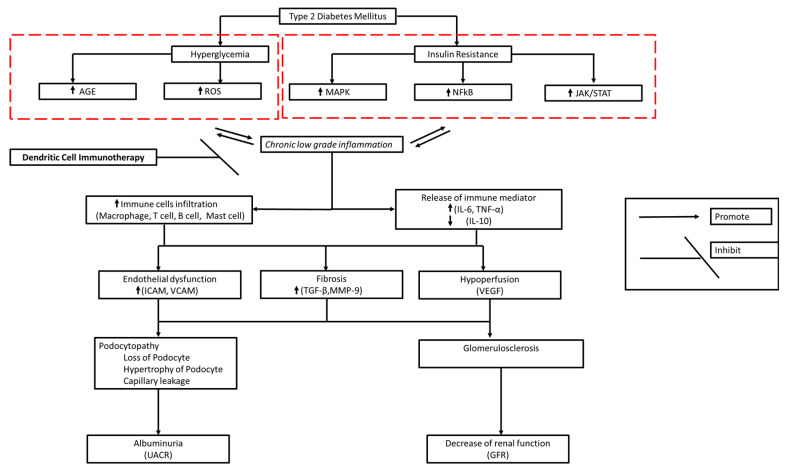
Proposed action mechanism of DC immunotherapy for DKD. Immunomodulation activity of DC can inhibit chronic low-grade inflammation, which is expected to stop renal damage and maintain kidney function.

## Data Availability

All data is available upon request from the authors.

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
