# Peer review of "The Potential of Anti-Inflammatory DC Immunotherapy in Improving Proteinuria in Type 2 Diabetes Mellitus"

_vaccines, 2024, doi:10.3390/vaccines12090972_

Round 1

Reviewer 1 Report

Comments and Suggestions for Authors

Current review manuscript discussed extensively about the role of dendritic cells in the modulation of inflammatory response and possible beneficial effects in the pathogenesis of diabetic kidney disease (DKD). There is lack of evidence suggesting the dendritic cells and their role in renal protective effects in the DKD progression. However, there is many evidences suggesting the involvement of DCs (cDC1) associated with worsening the disease progression under DKD. Also serval animal models of kidney disease in diabetes, indicating the activation of dendritic cell population and recruited or accumulated in kidney tissue in the diabetic environment. However, current manuscript could not show evidences suggesting the how the dendritic cell therapy can beneficial in preventing the DKD progression? Also authors did not describe kidney DCs and their subsets, and the role in the pathogenesis of DN.

Additional Comments:

Recent evidences suggesting that cGAS-sting pathway seems to play important role in the progression of DKD by modulating the inflammatory response and inhibition of sting pathway has shown to have protective effects in the progression of DKD. Authors need to include or discuss in Section 2, Diabetic kidney disease immunopathology along with references. Also, I wonder if there is any co-relation between the inhibition of sting pathway and Dendritic cell involvement in DKD? If yes, please discuss.

Author Response

Response to Reviewer 1 Comments 

1.    Summary           
Thank you very much for taking the time to review this manuscript. Please find the detailed responses below and the corresponding revisions/corrections highlighted in the re-submitted files.  

2.    Questions for General Evaluation     Reviewer’s Evaluation     Response     and Revisions 
Is the work a significant contribution to   the field?  
Is the work well organized and           comprehensively described?  
Is the work scientifically sound and not           misleading?  
Are there appropriate and adequate   references to related and previous  work?  
Is     the     English     used     correct     and           readable?       
3.     Point-by-point     response     to           
Comments     and     Suggestions     for 
Authors 
Comments 1: The current review manuscript discussed extensively the role of dendritic cells in the modulation of inflammatory response and possible beneficial effects in the pathogenesis of diabetic kidney disease (DKD). There is a lack of evidence suggesting the dendritic cells and their role in renal protective effects in the DKD progression. However, there is much evidence suggesting that the involvement of DCs (cDC1) is associated with worsening the disease progression under DKD. Also serval animal models of kidney disease in diabetes, indicating the activation of dendritic cell population and recruited or accumulated in kidney tissue in the diabetic environment. However, the current manuscript could not show evidence suggesting how dendritic cell therapy can be beneficial in preventing DKD progression. Also, the authors did not describe kidney DCs and their subsets and their role in the pathogenesis of DN. 

Response 1: Thank you very for your valuable input and suggestion. We agree that there are plenty of evidence showing DC's involvement in DKD progression. However, we would like to emphasize the type of DC which are responsible for inflammatory DC, which exacerbates immune-related damage in kidney structures. In our article, we would like to explain the potential use of anti-inflammatoric type of Dendritic Cells (otherwise known as Tolerogenic Dendritic Cells) in the context of DKD treatment. We believe we have already pointed out the mechanism in which anti-inflammatoric DC can inhibit the ongoing systemic inflammation affecting Diabetic patients, and then eventually protects kidney function. Therefore, to clarify the confusion we revise the tittle to “The Potential of Anti-inflammatory DC Immunotherapy in Improving Proteinuria in Type 2 Diabetes Mellitus”. We also emphasize this in our abstract which can be seen in Line 18 to Line 25. The type of DC mentioned throughout the article refers to anti-inflammatory/tolerogenic DC.  

Additionally, we have also describe DC and their subsets and their role in the pathogenesis of DKD in section 3 line 252 to line 298.  

Comments 2: Recent evidences suggesting that cGAS-sting pathway seems to play important role in the progression of DKD by modulating the inflammatory response and inhibition of sting pathway has shown to have protective effects in the progression of DKD. Authors need to include or discuss in Section 2, Diabetic kidney disease immunopathology along with references. Also, I wonder if there is any co-relation between the inhibition of sting pathway and Dendritic cell involvement in DKD? If yes, please discuss. 

Response 2: Agree. We have, accordingly add several paragraph explaining about CGASSTING pathway and their relation to Dendritic Cells in section 2 line 216-line 250. 

4.    Response to Comments on the Quality of English Language Point 1: 

5.    Additional clarifications 

Reviewer 2 Report

Comments and Suggestions for Authors

This paper details the role of the immune system in the pathophysiology of DKD and explores the mechanisms of DC-based therapy, highlighting the potential for new therapeutic approaches. DCs, as antigen-presenting cells, bridge innate and adaptive immunity and are capable of modulating inflammation, making them a promising treatment for inflammatory diseases such as DKD.

-Is there only one report of DC Immunotherapy being conducted on humans? If there are multiple reports, please present their progress and discuss why satisfactory results have not been obtained in Is it possible that diabetes medications could affect this immune system? 

-Is it possible that diabetes medications could affect this immune system? Some oral hypoglycemic agents are considered to potentially inhibit the progression of DKD, but do these medications exert such effects through modulation of the immune system?

Author Response

Response to Reviewer 2 Comments 

1.    Summary           
Thank you very much for taking the time to review this manuscript. Please find the detailed responses below and the corresponding revisions/corrections highlighted in the re-submitted files.  

2.    Questions for General Evaluation     Reviewer’s Evaluation     Response     and Revisions 
Is the work a significant contribution to   the field?  
Is the work well organized and           comprehensively described?  
Is the work scientifically sound and not           misleading?  
Are there appropriate and adequate   references to related and previous  work?  
Is     the     English     used     correct     and           readable?       
3.     Point-by-point     response     to           
Comments     and     Suggestions     for 
Authors 
Comments 1: This paper details the role of the immune system in the pathophysiology of DKD and explores the mechanisms of DC-based therapy, highlighting the potential for new therapeutic approaches. DCs, as antigen-presenting cells, bridge innate and adaptive immunity and are capable of modulating inflammation, making them a promising treatment for inflammatory diseases such as DKD. 
-Is there only one report of DC Immunotherapy being conducted on humans? If there are multiple reports, please present their progress and discuss why satisfactory results have not been obtained in Is it possible that diabetes medications could affect this immune system?  

Response 1: Thank you for pointing this out. We agree with this comment. Unfortunately, we can’t find report of DC immunotherapy in the context of diabetic kidney disease. We have highlighted this finding in Section 5 Paragraph 1 Line 383-386. We can only find one report of using autologous hematopoietic stem cells for treatment critical limb ischemia in Diabetic patients with Chronic Kidney Disease. Due to similar mechanism of action, which is by targeting inflammation, we have highlighted several studies regarding the use of autologous stem cells for treatment of Diabetic Kidney Disease, as well as discuss their progress in Section 5 line 381-444.    

Comments 2: -Is it possible that diabetes medications could affect this immune system? Some oral hypoglycemic agents are considered to potentially inhibit the progression of DKD, but do these medications exert such effects through modulation of the immune system? 

Response 2: We agree that several diabetes medications can inhibit the progression of DKD. We also find that such effects are also exterted through modulation of immune system. So that, we have include this topic in section 7, line 442 to line 491.  

4.    Response to Comments on the Quality of English Language Point 1: 

5.    Additional clarifications 
